# Incidence and Risk Factors for Notifiable Typhoid and Paratyphoid in Taiwan during the Period 2011–2020

**DOI:** 10.3390/healthcare9101316

**Published:** 2021-10-01

**Authors:** Fu-Huang Lin, Bao-Chung Chen, Yu-Ching Chou, Chi-Jeng Hsieh, Chia-Peng Yu

**Affiliations:** 1School of Public Health, National Defense Medical Center, Taipei City 11490, Taiwan; noldling@ms10.hinet.net (F.-H.L.); trishow@mail.ndmctsgh.edu.tw (Y.-C.C.); 2Division of Gastroenterology, Department of Internal Medicine, Tri-Service General Hospital, National Defense Medical Center, Taipei City 11490, Taiwan; staineely@yahoo.com.tw; 3Department of Health Care Administration, Asia Eastern University of Science and Technology, New Taipei City 22061, Taiwan; fl004@mail.aeust.edu.tw

**Keywords:** typhoid, paratyphoid, epidemiology, domestic, imported, Taiwan

## Abstract

The risk of the geographic transmission of emerging infectious diseases through air travel varies greatly. In this study, we collected data on cases of food-borne diseases between the years 2011 and 2020 in Taiwan to access the epidemiological features, differences, and trends in domestic and imported cases of typhoid and paratyphoid in terms of patient sex, age, month of confirmation, and area of residence. In this study, we made use of the open data website provided by Taiwan’s Centers for Disease Control (TCDC) to extract the reported numbers of cases of typhoid and paratyphoid between January and December from 2011 to 2020 for comparison. Univariate analysis was performed using the Chi-square test for categorical variables. Fisher’s exact test was performed if an expected frequency was less than 5. A total of 226 typhoid cases and 61 paratyphoid cases were analyzed from the database. The incidences of typhoid and paratyphoid per million of the population were 0.42–2.11 and 0–0.39, respectively. There was a significant difference in the incidence of the diseases between the age groups (*p* = 0.019), with a gradual increase in the 20–40 years group. A distinct seasonal (between fall and spring) variation was also observed (*p* = 0.012). There were 34 cases of children with typhoid in the period 2011–2015 and 12 cases of children with typhoid in the period 2016–2020. During these periods, there were two cases of paratyphoid. This study indicated that the risk of children suffering from typhoid has been significantly reduced in the last five years. Furthermore, we found that more women have acquired typhoid and paratyphoid than men, and that living in the Taipei metropolitan area and the northern area was a potential risk factor. Furthermore, the number of imported cases of typhoid (*n* = 3) and paratyphoid (*n* = 0) reported during the COVID-19 pandemic was lower than that reported for the same disease from 2011 to 2020. More typhoid and paratyphoid cases were imported from Indonesia, India, Myanmar, and Cambodia. This study represents the first report on confirmed cases of acquired typhoid and paratyphoid from surveillance data from Taiwan’s CDC for the period 2011–2020. This study also demonstrates that the cases of typhoid and paratyphoid decreased in Taiwan during the COVID pandemic. Big data were used in this study, which may inform future surveillance and research efforts in Taiwan.

## 1. Introduction

Typhoid and paratyphoid are both intestinal infectious diseases caused by the pathogenic bacteria *Salmonella enteric**a* serotype Typhi and *Salmonella enteric**a* serovar Paratyphi (A, B, and C), respectively [1]. Typhoid and paratyphoid are global diseases; however, they are most prevalent in developing countries, especially in areas where tap water is not widely available or environmental sanitation is poor [1]. People living in South Asia have the highest risk of infection [2]. In addition, people living in Southeast Asia, Africa, Central and South America, and the Caribbean islands also have a high risk of infection [3,4,5,6,7]. In recent years, the number of cases of typhoid and paratyphoid in developed countries has been drastically reduced due to improvements in sanitation.

Typhoid and paratyphoid are transmitted by ingesting food or drinking water contaminated by the feces or urine of patients or carriers [8]. Shellfish, fruits, and vegetables might all be contaminated and become infectious vectors, and the carriers may contaminate food when it is handled [9]. In addition, flies may spread bacteria in food and be one of the transmission routes [10].

The incubation period varies with the number of infectious bacteria. The incubation period of typhoid is typically 8 to 14 days (ranging from 3 to 60 days); the incubation period of paratyphoid is 1 to 10 days. After being infected with typhoid or paratyphoid, persistent fever, headache, malaise, anorexia, abdominal pain, relative slowing of heart rhythm, spleen enlargement, rash, cough, constipation or diarrhea, and lymphatic enlargement may occur [11,12]. If typhoid is not treated, bleeding can also be observed in more serious cases [13]. Paratyphoid has milder symptoms and a lower mortality rate than typhoid. A patient with typhoid or paratyphoid might become a carrier in the acute phase in mild case, or after asymptomatic infection [14].

In Taiwan, both typhoid and paratyphoid have been listed as notifiable diseases since 1944 [15]. Typhoid has largely been controlled by effective vaccinations and a strengthened surveillance system. There have still been some reports of cases, both imported and locally transmitted, and most imported cases contract the disease in China and Southeast Asia. Additionally, to the best of our knowledge, no research has provided longitudinal epidemiological data on typhoid and paratyphoid in Taiwan. Thus, we analyzed cases of food-borne infectious diseases between 2011 and 2020 in Taiwan to access the impact of COVID-19 on the epidemiological features of typhoid and paratyphoid. In addition, we compared the distribution of these diseases both before and after the COVID-19 outbreak in this study.

## 2. Materials and Methods

### 2.1. Ethical Policy

In this study, we used information that is freely available in the public domain and analyzed open data sets where data have been properly anonymized; therefore, this study did not require ethical approval [16,17].

### 2.2. Data Source

In Taiwan, notifiable diseases must be reported to the Centers for Disease Control (CDC). Taiwan National Infectious Disease Statistics System (TNIDSS) has an open data website which includes data about such diseases. For this study, information about cases of typhoid and paratyphoid between January and December from 2011 to 2020 was taken from this website [18].

### 2.3. Data Analysis

This is a retrospective historical study of all typhoid and paratyphoid cases since 2011. We confirmed the number of people diagnosed as having typhoid and paratyphoid from 2011 to 2020 and examined the distribution of their epidemiological characteristics (sex, age, time of diagnosis, living area), differences, and results. Descriptive data are shown as mean and summary statistics, where appropriate. Categorical variables were compared using the chi-square test. All statistical analyses were performed using SPSS (IBM SPSS version 21; Asia Analytics Taiwan Ltd., Taipei, Taiwan). All statistical tests were 2-sided with α value of 0.05, and *p* values < 0.05 were considered to represent statistical significance.

## 3. Results

In this study, we investigated the epidemiological features of domestic and imported cases of typhoid and paratyphoid in Taiwan during the period 2011–2020. The total number of typhoid cases was 226, the total number of domestic cases was 107, and the total number of imported cases was 119 (Table 1). For paratyphoid, the total number of cases was 61, the total number of domestic cases was 19, and the total number of imported cases was 42 (Table 2). The epidemiological features found by examining the typhoid and paratyphoid cases are shown in Table 3. Among the 109 imported cases for which the region the typhoid was contracted in was known, 58 (53.2%) had traveled to Indonesia, 15 (13.8%) had traveled to India, 14 (12.8%) had traveled to Myanmar, 13 (11.9%) had traveled to the Philippines, 3 (2.8%) had traveled to Malaysia, 3 (2.8%) had traveled to Cambodia, and 3 (2.8%) had traveled to China. For another 10 imported typhoid cases, the data on the region where the disease was contracted are not available. For travelers from Taiwan, the risk ratio (RR) of contracting typhoid was 304.3 for travel to Indonesia, 473.6 for travel to India, 1349.5 for travel to Myanmar, 57.4 for travel to the Philippines, 6.7 for travel to Malaysia, and 533.0 for travel to Cambodia compared with the risk for travel to China (Table 4). Among the 39 imported cases for which the region the paratyphoid was contracted in was known, 15 (38.5%) had traveled to Indonesia, 10 (25.6%) had traveled to India, 7 (17.9%) had traveled to Cambodia, 3 (7.7%) had traveled to Myanmar, 2 (5.1%) had traveled to Bangladesh and 2 (5.1%) had traveled to China. For another 3 imported typhoid cases, the data on the region where the disease was contracted are not available. For travelers from Taiwan, the RR of contracting paratyphoid was 118.1 for travel to Indonesia, 473.6 for travel to India, 1865.4 for travel to Cambodia, 433.8 for travel to Myanmar, and 2718.2 for travel to Bangladesh compared with the risk for travel to China (Table 4). 

Three cases of typhoid were reported in 2020, which was lower than that in 2019 (*n* = 17), 2018 (*n* = 13), 2017 (*n* = 13), 2016 (*n* = 9), 2015 (*n* = 14), 2014 (*n* = 19), 2013 (*n* = 13), 2012 (*n* = 11), and 2011 (*n* = 7) (Figure 1). Zero cases of paratyphoid were reported in 2020, which was lower than that in 2019 (*n* = 6), 2018 (*n* = 7), 2017 (*n* = 3), 2016 (*n* = 3), 2015 (*n* = 1), 2014 (*n* = 8), 2013 (*n* = 2), 2012 (*n* = 7), and 2011 (*n* = 5) (Figure 2). The typhoid and paratyphoid cases were imported to Taiwan from five continents (Figure 3) and various countries (Figure 4).

## 4. Discussion

The current era of quick and convenient international travel and recent changes in the global climate have increased the risk of contracting tropical diseases, including typhoid and paratyphoid, in Taiwan, and the control of these diseases is an ever-present challenge. In 2017, an estimated 14 million cases of enteric fever occurred worldwide, resulting in about 136,000 deaths [19]. More than 80% of these cases occurred in South and Southeast Asia and in Sub-Saharan Africa. We researched the annual summary data of imported and domestic typhoid and paratyphoid cases published by the Taiwan CDC during the period 2011–2020. To the best of our knowledge, this is the first report in Taiwan in the past ten years to investigate the incidence of typhoid and paratyphoid in Taiwan. During the period of investigation, more than 90% of the cases were imported from Asian countries. Between January 2011 and December 2020, imported cases reached a cumulative total of 161 (119 for typhoid and 42 for paratyphoid). Most of the imported cases originated in Indonesia. Hotspots for typhoid and paratyphoid are primarily in the tropical and subtropical areas of the Northern Hemisphere. In most cases, the disease is spread in areas with a high temperature and a large numbers of travelers. Air transport-related services are an important cause of the transmission of zoonoses (e.g., typhoid and paratyphoid), and lively airports are hotbeds for pathogens (viruses and bacteria, etc.). To maintain effective epidemic control, governments have a responsibility to offer information on limiting the spread of diseases to travelers arriving from disease hotspots and establish control measures. Our study proposes that the incidence of endemic enteric fever has decreased greatly in developed countries (e.g., Taiwan), likely because of the widespread availability of safe food and water in recent years.

This study compared the number of typhoid and paratyphoid cases from 2011 to 2020. Sex was not found to be a risk factor for enteric fever; nevertheless, the attack rate was found to be a little greater in women (58.4% for typhoid cases and 63.9% for paratyphoid cases) than in men. A previous study [20] indicated that the sex-specific burden of salmonellosis varies by age (20–39 years female-to-male incidence rate ratio of 1.09, 40–59 years female-to-male incidence rate ratio of 1.23, and over 60 years female-to-male incidence rate ratio of 1.08), with the result being similar to that of this study. There was, however, a significant difference in the age of patients. Both typhoid cases and paratyphoid cases primarily occurred in individuals who were 20–39 years old (typhoid attack rate = 54.4%; paratyphoid attack rate = 59.0%). A previous study [21] indicated that there was a higher typhoid prevalence in adults older than 18 years, with the result being similar to that found in this study. The reason for this was that young adults were more likely to visit areas in which the virus was foodborne and waterborne and thus they were infected more frequently than other age groups; it is also possible that people in this age group may be more likely to underrate the risk. There was also a significant difference in the season in which the diseases were contracted; typhoid cases tended to rise in the fall and winter, whereas paratyphoid cases increased in the spring and summer. A previous study [22] found that the greatest incidence of diarrhea and typhoid occurred in the summer; this result is not similar to that of this study. This study suggested that the different prevalences of typhoid in the different seasons may have occurred due to the differences in hygiene or culture between Taiwan and Afghanistan. This may be because imported cases enhanced the spread of the disease, thereby indirectly increasing the number of domestic cases. Furthermore, no significant difference was observed in the cases according to individuals’ places of residence. Taipei is a metropolitan center in which residents are probably more likely to come into contact with imported cases; thus, they had a higher risk of acquiring the diseases. Overall, the differences between the typhoid and paratyphoid cases in terms of patient age and the season of confirmation may allow us to find risk factors for enteric fever in the Taiwanese population.

Research in other countries has shown that hot environments are associated with rapid disease transmission on account of contamination with pathogens [23]. In its annual assessment report in late 2019, the United Nations World Meteorological Organization reported that climate change has exceeded humans’ ability to adapt to growing temperatures every decade for the past 40 years [24]. The global temperature has grown by 1.1 °C since the industrial revolution, and this decade will be the hotter than the one before. The average temperature in Taiwan was 24.56 °C in 2019, which was the highest since 1947 [25]. Bearing this in mind, the confirmed cases of typhoid and paratyphoid in the fall of 2019 were likely due to the hotter temperatures during the year. The rise in temperature will enhance bacterial reproduction in the community and help the disease to spread widely. Fall and winter have become hotter because of global warming; this has likely added to the dissemination of foodborne diseases (for example, typhoid and paratyphoid), posing a grave threat to human health. Under such circumstances, the government is urged to undertake advanced planning of public-health policies and implement preventive measures to safeguard the local population.

In a recent article, de Miguel Buckley et al. [26] reported that in Spain, the transmission of campylobacteriosis had seemed to be suppressed during the COVID-19 pandemic. A decline in foodborne diseases such as salmonellosis was also reported during early 2020 in Spain [26]; this study shows a similar scenario for salmonellosis (typhoid and paratyphoid) in Taiwan. The number of typhoid cases was reduced to three and the number of paratyphoid cases dramatically reduced to zero in 2020, which marks the beginning of the COVID-19 outbreak. The level of zero cases in a year has not been achieved in the previous nine years. The reason for the decline of imported cases of typhoid and paratyphoid (transmission) in Taiwan may be the implementation of aggressive infection control measures during this time. “Border control”, particularly concerning China, was implemented early in the COVID-19 outbreak in Wuhan [27] and may have resulted in a lower number of imported typhoid cases and no paratyphoid cases, as well as indirectly decreasing the occurrence of imported infection from other countries where local transmission takes place. Furthermore, infection prevention practices, such as wearing masks, dead areas, and social distancing might have helped to reduce the risk of the typhoid/paratyphoid spreading through the infectious food/water route by contact, drinking, or eating. These interventions may have prevented the circulation of locally transmitted typhoid/paratyphoid cases, similar to other foodborne infectious diseases in Italy [28].

Data from other studies show that approximately half of travelers with typhoid and paratyphoid returning to developed countries came from Asia [29]. Our study showed similar findings for Taiwan. Previous studies have indicated that the incidence among children between 2 and 4 years of age and young adults is high on the Indian subcontinent [30,31,32]. Immigration and the increased movement of international populations (e.g., travelers from Indonesia and India) have affected the morbidity of this disease in Taiwan, mirroring the findings of a previous study [33]. In addition, this study found that the number of cases of typhoid or paratyphoid among patients increased when the number of imported cases of typhoid or paratyphoid among immigrants and international travelers increased. This study suggests that international passengers arriving from abroad and going to a destination country might be one of the risk factors for contracting typhoid and paratyphoid in Taiwan.

However, it is possible that under reporting or under diagnosis occurred due to the fear of visiting hospitals during the COVID-19 pandemic, which may have resulted in an underestimation of the actual number of typhoid and paratyphoid cases. Further studies comparing the number of people visiting clinics for diarrheal symptoms during the COVID-19 pandemic with those in previous years may help us to evaluate the prevalence of this issue. 

There were two limitations in the present study. One is that the Taiwan CDC’s Taiwan National Infectious Disease Statistics System (TNIDSS) includes basic epidemiological data for patients with typhoid and paratyphoid but provides no clinical data. It is not possible to compare clinical data between patients in terms of their differences or trends. Second, the data provided by TNIDSS contain no information about the genotypes or strains of the *salmonella enterica* serotype Typhi or *salmonella enterica* serovar Paratyphi isolated. Therefore, (1) the type of *salmonella enterica* serotype Typhi and *salmonella enterica* serovar Paratyphi strain that spread to Taiwan and (2) the affinity between bacterial strains in Taiwan and other countries were not analyzed in this study. Nonetheless, the advantage offered by this study was access to the diverse data provided by Taiwan’s public sector on its online platform (including the initial version of platform) and the evaluation of the impact of COVID-19 on the epidemiological features of typhoid and paratyphoid.

## 5. Conclusions

In this study, the epidemiological characteristics and trends of imported and domestic cases of typhoid and paratyphoid in Taiwan from 2011 to 2020 were investigated. Data from the Taiwan CDC open database indicated that there were 226 cases of typhoid and 61 cases of paratyphoid over the period studied. There were significant differences in age (most cases occurred in individuals between 20 and 40 years old), season of confirmation (fall for typhoid and spring for paratyphoid accounted for the highest proportions of cases), and place of residence (Taipei city area and the northern area accounted for the largest proportions) in the data. In this study, we also demonstrated that typhoid and paratyphoid showed a decreased transmission in Taiwan during the COVID-19 pandemic. This study highlights the importance of longitudinal and geographically extended studies for understanding the implications of zoonotic disease transmission in the Taiwanese population. 

## Figures and Tables

**Figure 1 healthcare-09-01316-f001:**
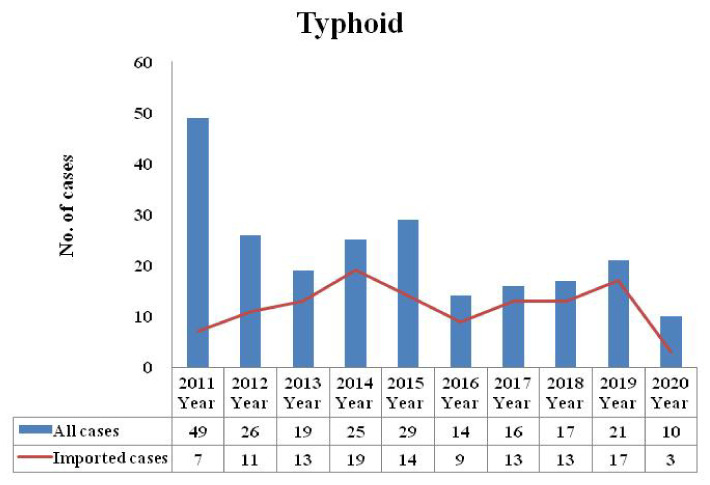
Number of typhoid cases reported in Taiwan from 2011 to 2020 (total (blue) and imported (red)). The total number of typhoid cases will increase due to the increase in the number of imported cases.

**Figure 2 healthcare-09-01316-f002:**
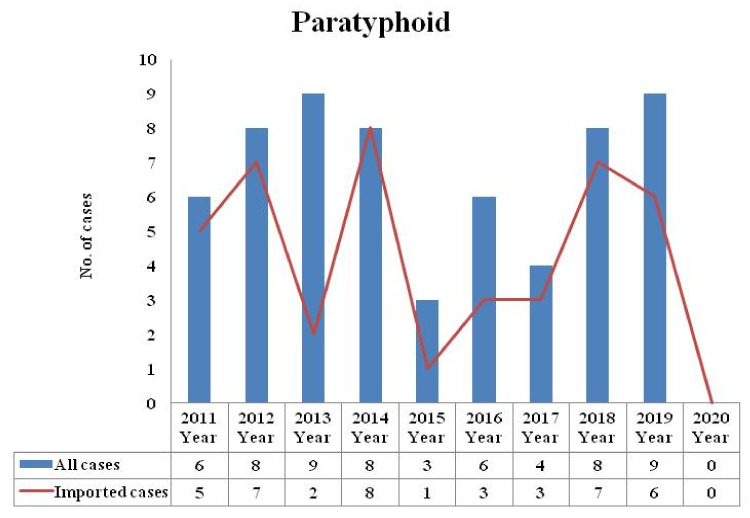
Number of paratyphoid cases reported in Taiwan from 2011 to 2020 (total (blue) and imported (red)). The total number of paratyphoid cases will increase due to the increase in the number of imported cases.

**Figure 3 healthcare-09-01316-f003:**
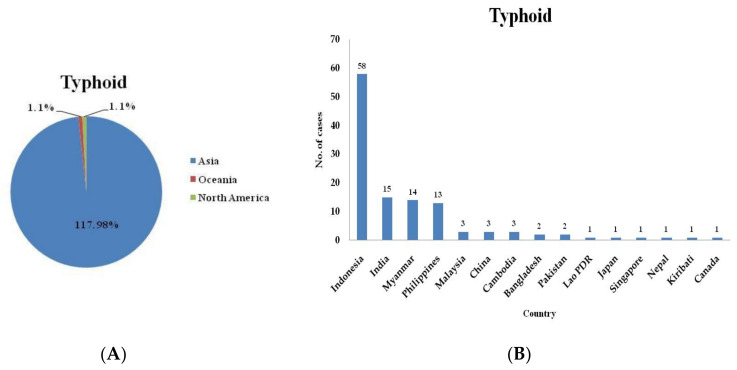
Typhoid cases reported in Taiwan arriving from Asian countries and other countries (**A**) and cases imported from Indonesia and other countries (**B**) from 2011 to 2020. Indonesia (in Asia) is the country in which the most typhoid cases imported into Taiwan originated.

**Figure 4 healthcare-09-01316-f004:**
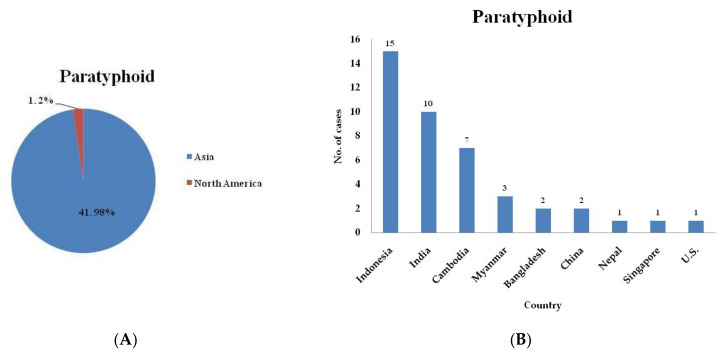
Paratyphoid cases reported in Taiwan arriving from Asian countries and other countries (**A**) and cases imported from Indonesia and other countries (**B**) from 2011 to 2020. Indonesia (in Asia) is the country in which the most typhoid cases imported into Taiwan originated.

**Table 1 healthcare-09-01316-t001:** Epidemiological features of domestic and imported cases of typhoid in Taiwan during the period 2011–2020.

Variables	All Cases		Domestic Cases		Imported Cases	
	*n* = 226	%	*n* = 107	%	*n* = 119	%
Sex						
Male	94	41.6	48	44.9	46	38.6
Female	132	58.4	59	55.1	73	61.4
Age group						
<20	46	20.3	24	22.4	22	18.5
20–39	123	54.4	47	43.9	76	63.9
40–59	32	14.2	15	14.0	17	14.3
≥60	25	11.1	21	19.6	4	3.3
Year group						
2011–2015	148	65.5	84	78.5	64	53.8
2016–2020	78	34.5	23	21.5	55	46.2
Season						
Spring	51	22.6	23	21.5	28	23.5
Summer	51	22.6	19	17.8	32	26.9
Fall	64	28.3	41	38.3	23	19.3
Winter	60	26.5	24	22.4	36	30.3
Residency						
Northern	155	68.6	76	71.0	79	66.4
Central	27	11.9	10	9.3	17	14.3
Southern	42	18.6	20	18.7	22	18.5
Eastern	2	0.9	1	1.0	1	0.8

**Table 2 healthcare-09-01316-t002:** Epidemiological features of domestic and imported cases of paratyphoid in Taiwan during the period 2011–2020.

Variables	All Cases		Domestic Cases		Imported Cases	
	*n* = 61	%	*n* = 19	%	*n* = 42	%
Sex						
Male	22	36.1	1	5.3	21	50
Female	39	63.9	18	94.7	21	50
Age group						
<20	4	6.6	0	0	4	9.5
20–39	36	59.0	10	52.6	26	61.9
40–59	16	26.2	6	31.6	10	23.8
≥60	5	8.2	3	15.8	2	4.8
Year group						
2011–2015	34	55.7	11	57.9	23	54.8
2016–2020	27	44.3	8	42.1	19	45.2
Season						
Spring	22	36.1	8	42.1	14	33.3
Summer	20	32.8	7	36.8	13	31.0
Fall	10	16.4	2	10.5	8	19.0
Winter	9	14.7	2	10.5	7	16.7
Residency						
Northern	39	63.9	12	63.2	27	64.3
Central	15	24.6	5	26.3	10	23.8
Southern	7	11.5	2	10.5	5	11.9
Eastern	0	0	0	0	0	0

**Table 3 healthcare-09-01316-t003:** Epidemiological features of typhoid and paratyphoid cases in Taiwan during the period 2011–2020.

Variables	Typhoid Cases		Paratyphoid Cases		*p*
	*n* = 226	%	*n* = 61	%	
Sex					
Male	94	41.6	22	36.1	0.435
Female	132	58.4	39	63.9	
Age group					
<20	46	20.3	4	6.6	0.019
20–39	123	54.4	36	59.0	
40–59	32	14.2	16	26.2	
≥60	25	11.1	5	8.2	
Year group					
2011–2015	148	65.5	34	55.7	0.161
2016–2020	78	34.5	27	44.3	
Season					
Spring	51	22.6	22	36.1	0.012
Summer	51	22.6	20	32.8	
Fall	64	28.3	10	16.4	
Winter	60	26.5	9	14.7	
Residency					
Northern	155	68.6	39	63.9	0.061
Central	27	11.9	15	24.6	
Southern	42	18.6	7	11.5	
Eastern	2	0.9	0	0	

**Table 4 healthcare-09-01316-t004:** Travel destinations of 109 imported cases of typhoid and 39 imported cases of paratyphoid in Taiwan between 2011 and 2020.

Country of Destination	No. Cases (%)	No. of Air Passengers (100,000)	RR
Typhoid			
China	3	271.82	Reference
Indonesia	58	17.27	304.30
India	15	2.87	473.55
Myanmar	14	0.94	1349.46
Philippines	13	20.52	57.40
Malaysia	3	40.53	6.71
Cambodia	3	0.51	532.98
Paratyphoid			
China	2	271.82	Reference
Indonesia	15	17.27	118.05
India	10	2.87	473.55
Cambodia	7	0.51	1865.43
Myanmar	3	0.94	433.76
Bangladesh	2	0.10	2718.2

Note: Only countries with at least three cases of typhoid and at least two cases of paratyphoid are listed.

## Data Availability

Taiwan Centers for Disease Control. Taiwan National Infectious Disease Statistics System. Available online: https://nidss.cdc.gov.tw/ch/ (accessed on 1 July 2021).

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
