# Peer review of "Incidence and Risk Factors for Notifiable Typhoid and Paratyphoid in Taiwan during the Period 2011–2020"

_healthcare, 2021, doi:10.3390/healthcare9101316_

Round 1

Reviewer 1 Report

General Comments

The manuscript requires considerale language editing.

The abstract needs to be revised, especially in the the sentec structure. There seem to be too many repetitions making the abstract unnecessarily lengthy (Some examples are provided under specific comments). There is also no mention of what analyses were performed. The abstract should be undeestood as a standalone piece. What parameters were investigated?

The result section needs to be improved. There is repetition of information in some case. The authors repeat in ther text what is found in the figures.The overall quality of the figures (1-3) need to be improved.

Specific comments

Line 16: Delete “was”

Line 18-19: Change “This study analyzed 226 cases with typhoid and 61 cases with paratyphoid from Taiwan’s CDC databases” to “A total of 226 typhoid cases, and 61 paratyphoid cases were analyzed from the database”

Line 20: Change “is” to “was”

Line 20-23: “In this study….respectively”. Change to “There was a significant difference in the incidence of the diseases between the age groups (p = 0.019), gradual increase in the 20-40 years group. A distinct seasonal (between Fall and Spring) variation was also observed (p = 0.012).

Line 40: Please check the nomenclature of Salmonella and correct where appropriate.

Line 41-42: ..global diseases” already means that they are distributed all over the world. So, the consitnuation of that sentence “with cases…world” is irrelevant. Please delete.

Line 44: “ Asia has the highest risk of the infection.” Please provide a reference.

Line 48: Delete “equipment”

Line 53-54: …and lymphatic enlargement may occur.

Line 56-62: Reference

Line 69-73: This section should come just before 49-56

Line 78: We analysed cases of…..

Figure 1 to 3: These figures need to be improved. Also, the textpreceding these Figures is a repetition of the Figures. Please just point the reader to the key findings on each figure.

Line 147: internationalel???

Line 156-157: Delete “and were more than the number of local cases”

Line 160: Most…

Line 169-170: Delete “It showed no significant difference in the”

Line 170: in particular

Line 175: foodborne. (Also be consistent with this spelling throughout the manuscript)

Author Response

Dear the reviewers,                                                  September 14, 2021

  We resubmitted the manuscript entitled “Incidence and risk factors of notifiable typhoid and paratyphoid in Taiwan during 2011-2020” to the Journal after amendments made based on reviewers comments. We have carefully revised our manuscript according to reviewers’ critiques and suggestions. We marked amendments in yellow font in the manuscript for clarity purpose. Our specific responses to reviewers’ comments are as follows.

Reviewer 1

General Comments

  1. The manuscript requires considerale language editing.

Response: Thanks the reviewer comment. The authors have attached the language editing.

  1. The abstract needs to be revised, especially in the the sentec structure. There seem to be too many repetitions making the abstract unnecessarily lengthy (Some examples are provided under specific comments). There is also no mention of what analyses were performed. The abstract should be undeestood as a standalone piece. What parameters were investigated?

Response: The authors have agreed the reviewer comment. The authors have revised the sentence as line 14-25 in revised manuscript.

  1. The result section needs to be improved. There is repetition of information in some case. The authors repeat in ther text what is found in the figures. The overall quality of the figures (1-3) need to be improved.

Response: The authors have revised the sentence as line 97-110 in revised manuscript. The figures (1-4) have improved as line 145-157 in revised manuscript.

 Specific comments

  1. Line 16: Delete “was”

Response: Thanks the reviewer comment. The authors have deleted “was” as line 17 revised manuscript.

  1. Line 18-19: Change “This study analyzed 226 cases with typhoid and 61 cases with paratyphoid from Taiwan’s CDC databases” to “A total of 226 typhoid cases, and 61 paratyphoid cases were analyzed from the database”

Response: The authors have revised the sentence as line 21-22 in revised manuscript.

  1. Line 20: Change “is” to “was”

Response: The authors have revised the word as line 23 in revised manuscript.

  1. Line 20-23: “In this study….respectively”. Change to “There was a significant difference in the incidence of the diseases between the age groups (= 0.019), gradual increase in the 20-40 years group. A distinct seasonal (between Fall and Spring) variation was also observed (= 0.012).

 Response: The authors have revised the sentence as line 23-25 in revised manuscript.

  1. Line 40: Please check the nomenclature of Salmonella and correct where appropriate.

Response: The authors have revised the sentence as line 43-44 in revised manuscript.

  1. Line 41-42: ..global diseases” already means that they are distributed all over the world. So, the consitnuation of that sentence “with cases…world” is irrelevant. Please delete.

Response: The authors have revised the sentence as line 44 in revised manuscript.

  1. Line 44: “ Asia has the highest risk of the infection.” Please provide a reference.

Response: The authors have added the reference number 2 as line 299-300 in revised manuscript.

  1. Line 48: Delete “equipment”

Response: The authors have deleted “equipment” as line 50 revised manuscript.

Line 53-54: …and lymphatic enlargement may occur.

Response: The authors have revised the sentence as line 60-61 in revised manuscript.

Line 56-62: Reference

Response: Because the other reviewer comments, the authors have deleted the paragraph/ sentences as line 64 in revised manuscript.

Line 69-73: This section should come just before 49-56

Response: The authors have revised the sentence as line 51-55 in revised manuscript.

Line 78: We analysed cases of…..

Response: The authors have revised the sentence as line 70 in revised manuscript.

Figure 1 to 3: These figures need to be improved. Also, the textpreceding these Figures is a repetition of the Figures. Please just point the reader to the key findings on each figure.

Response: The authors have improved figure 1-4 as line 145-157 in revised manuscript.

Line 147: internationalel???

Response: The authors have revised the word as line 159 in revised manuscript.

Line 156-157: Delete “and were more than the number of local cases”

Response: The authors have deleted the sentence as line 168 in revised manuscript.

Line 160: Most…

Response: The authors have revised the word as line 170 in revised manuscript.

Line 169-170: Delete “It showed no significant difference in the”

Response: The authors have deleted the sentence as line 181-182 in revised manuscript.

Line 170: in particular

Response: The authors have deleted the word as line 181-182 in revised manuscript.

Line 175: foodborne. (Also be consistent with this spelling throughout the manuscript)

Response: The authors have deleted the word as line 193, also be consistent with this spelling throughout the revised manuscript.

Hopefully, our revised manuscript could fulfill your scientific requirements for publication.

Sincerely yours,

Chia-Peng Yu, Ph.D. (the corresponding author)

School of Public Health,

National Defense Medical Center

No.161 Sec. 6, Minquan E. Rd., Neihu Dist., Taipei 114, Taiwan, Republic of China,

Tel: +886-2-87923311 ext. 16791, Fax: +886-2-87924379,

Reviewer 2 Report

The study is of local interest. In addition, the findings are poorly presented (mostly by showing tables) and discussion with already existent literature/comparison with similar findings from other countries is largely missing.

General comments:

The introduction contains several pieces of information unrelated to the present study. I propose to remove paragraphs/sentences associated with general knowledge and pathogenesis of typhoid/paratyphoid, as they distract the reader from following this specific work. 

The aim of the present study is not being provided in the last paragraph of the introduction. It is not clear to me what is missing from the literature and which gap is this study filling. 

Discussion is also extended for this study. There are whole paragraphs where the authors cite one reference only, e.g. lines 201-218., or even none, for example, lines 169-185. Rather than re-presenting results, a discussion with similar findings of the literature is important. 

Conclusions are very lengthy, almost 20 lines. This paragraph should be reduced by over 50% to provide only the critical results of the study, not a repetition of the whole introduction, or unrelated information, such as COVID-19 stuff, as the period is short and the case data are limited. 

Specific comments:

Line 14. remove "infectious" after "food-borne".

Lines 16-17. Using "CDC" may be confusing (CDC of USA). Consider using TCDC instead. 

Line 22 versus line 29 (and elsewhere in text): choose a uniform style for using the "=" symbol, i.e. with or without spaces before and after it, and not interchangeably. 

Line 36 Taiwan should be used as a key-word.

Line 40. the correct nomenclature is "Salmonella enterica" (not enteric) which should be italicized also. 

Line 51. "After being infected with typhoid or paratyphoid fever".... People are being infected with the respective bacteria, not with "fever". Please rephrase accordingly. This should be done throughout the text.

Lines 54-56. Please provide a relevant reference. Consider rephrasing the sentence which contains "it may cause bleeding or perforation of the small intestine" to a more indirect phrase (e.g. bleeding can also be observed in more serious cases... etc) according to the literature.

Lines 57-62. Please provide references. Consider my previous comment for rephrasing "infection with typhoid and paratyphoid fever". 

Lines 64-68. Please provide references.

Lines 70-73. References are required. Foodstuffs are not considered vectors, this applies only to living organisms. 

Line 78. what do the authors mean by "we conducted data" ??

Lines 86-87. This is unrelated to ethics. 

Line 105 (and elsewhere): rephrase "Table 1 showed". It is not a past condition. Additionally, do not just show the tables. a little information on the numbers is important, e.g. number of total cases, number of imported cases, etc. 

Line 108: 109 imported cases for typhoid for which the region was known (shown also in Table 4), Table 1 shows 119 imported cases. Are they the same (and there is a typo) or there are 10 cases for which the region was not known? This also applies for paratyphoid, 42 imported cases in Table 2, 39 imported cases in Table 4, and line 114. This is confusing and needs clarification.

Line 111: "RR" is risk ratio? Please write out in full.

Lines 111-112 (and elsewhere). "The RR of becoming typhoid fever". This gives the impression that the travelers turned into a fever! Keeping it simple (of getting typhoid fever) is more proper in such cases.

Lines 137-141. Use "total" rather than "domestic and imported" to discriminate it from "imported". 

Figures 3 and 4: The part B of the figures should be similarly presented. x-axis description of Fig.3B is inclined. The legend "imported cases" is redundant, as all of these are imported. Figure legends indicate "various continents" but this is misleading. Preferably "from Asian and other countries", or something similar. 

Line 170. "The gender" is used in the middle of a sentence.

Lines 202-203. campylobacteriosis and salmonellosis should not start with a capital letter. 

Author Response

Dear the reviewers,                                                  September 14, 2021

  We resubmitted the manuscript entitled “Incidence and risk factors of notifiable typhoid and paratyphoid in Taiwan during 2011-2020” to the Journal after amendments made based on reviewers comments. We have carefully revised our manuscript according to reviewers’ critiques and suggestions. We marked amendments in yellow font in the manuscript for clarity purpose. Our specific responses to reviewers’ comments are as follows.

Reviewer 2

The study is of local interest. In addition, the findings are poorly presented (mostly by showing tables) and discussion with already existent literature/comparison with similar findings from other countries is largely missing.

Response: Thanks the reviewer comment. The authors have revised the paragraph of discussion, add references which comparison with similar findings from other countries. See line 181-241 in the revised manuscript.

General comments:

  1. The introduction contains several pieces of information unrelated to the present study. I propose to remove paragraphs/sentences associated with general knowledge and pathogenesis of typhoid/paratyphoid, as they distract the reader from following this specific work. 

Response: Thanks the reviewer comment. The authors have deleted the 2 paragraphs of introduction. See line 51-64 in the revised manuscript.

  1. The aim of the present study is not being provided in the last paragraph of the introduction. It is not clear to me what is missing from the literature and which gap is this study filling. 

Response: Thanks the reviewer comment. The authors have added the sentence. See line 69-70 in the revised manuscript.

  1. Discussion is also extended for this study. There are whole paragraphs where the authors cite one reference only, e.g. lines 201-218., or even none, for example, lines 169-185. Rather than re-presenting results, a discussion with similar findings of the literature is important. 

Response: Thanks the reviewer comment. The authors have added the 5 references of discussion, no.20-22 and 27-28. See line 181-241 in the revised manuscript.

  1. Conclusions are very lengthy, almost 20 lines. This paragraph should be reduced by over 50% to provide only the critical results of the study, not a repetition of the whole introduction, or unrelated information, such as COVID-19 stuff, as the period is short and the case data are limited. 

Response: Thanks the reviewer comment. The authors have deleted over 50% for conclusions. See line 274-285 in the revised manuscript.

Specific comments:

  1. Line 14. remove "infectious" after "food-borne".

Response: The authors have revised the word as line 14 in revised manuscript.

  1. Lines 16-17. Using "CDC" may be confusing (CDC of USA). Consider using TCDC instead. 

Response: The authors have revised the word as line 18 in revised manuscript.

  1. Line 22 versus line 29 (and elsewhere in text): choose a uniform style for using the "=" symbol, i.e. with or without spaces before and after it, and not interchangeably. 

Response: The authors have choose a uniform style as line 24-32 in revised manuscript.

  1. Line 36 Taiwan should be used as a key-word.

Response: The authors have added the word as line 39 in revised manuscript.

  1. Line 40. the correct nomenclature is "Salmonella enterica" (not enteric) which should be italicized also. 

Response: The authors have added the sentence as line 43 in revised manuscript.

  1. Line 51. "After being infected with typhoid or paratyphoid fever".... People are being infected with the respective bacteria, not with "fever". Please rephrase accordingly. This should be done throughout the text.

Response: Thanks reviewer comments. The authors have deleted the words, not with "fever", so typhoid or paratyphoid be done throughout the revised manuscript.

  1. Lines 54-56. Please provide a relevant reference. Consider rephrasing the sentence which contains "it may cause bleeding or perforation of the small intestine" to a more indirect phrase (e.g. bleeding can also be observed in more serious cases... etc) according to the literature.

Response: Thanks reviewer comments. The authors have revised deleted the sentence” bleeding can also be observed in more serious cases”, and added reference [13]. See line 61-62 in the revised manuscript.

  1. Lines 57-62. Please provide references. Consider my previous comment for rephrasing "infection with typhoid and paratyphoid fever". 

Response: Thanks reviewer comments which remove paragraphs/sentences associated with general knowledge and pathogenesis of typhoid/paratyphoid. So, the authors have deleted the sentence. See line 51-64 in the revised manuscript.

  1. Lines 64-68. Please provide references.

Response: Thanks reviewer comments which remove paragraphs/sentences associated with general knowledge and pathogenesis of typhoid/paratyphoid. So, the authors have deleted the sentence. See line 51-64 in the revised manuscript.

  1. Lines 70-73. References are required. Foodstuffs are not considered vectors, this applies only to living organisms. 

Response: The authors have added two references. See line 51-55 in the revised manuscript.

  1. Line 78. what do the authors mean by "we conducted data" ??

Response: The authors have revised the sentence as line 70 in the revised manuscript.

  1. Lines 86-87. This is unrelated to ethics. 

Response: The authors have deleted the sentence which is unrelated to ethics. See line 78-79 in the revised manuscript.

  1. Line 105 (and elsewhere): rephrase "Table 1 showed". It is not a past condition. Additionally, do not just show the tables. a little information on the numbers is important, e.g. number of total cases, number of imported cases, etc. 

Response: The authors have revised the sentences as line 100-103 in the revised manuscript.

  1. Line 108: 109 imported cases for typhoid for which the region was known (shown also in Table 4), Table 1 shows 119 imported cases. Are they the same (and there is a typo) or there are 10 cases for which the region was not known? This also applies for paratyphoid, 42 imported cases in Table 2, 39 imported cases in Table 4, and line 114. This is confusing and needs clarification.

Response: The authors have revised the sentences as line 98-110 in the revised manuscript.

  1. Line 111: "RR" is risk ratio? Please write out in full.

Response: The authors have revised the words as line 110 in the revised manuscript.

  1. Lines 111-112 (and elsewhere). "The RR of becoming typhoid fever". This gives the impression that the travelers turned into a fever! Keeping it simple (of getting typhoid fever) is more proper in such cases.

Response: The authors have replaced “typhoid and paratyphoid fever” with “typhoid and paratyphoid”, not with “fever” in the revised manuscript.

  1. Lines 137-141. Use "total" rather than "domestic and imported" to discriminate it from "imported". 

Response: The authors have replaced “domestic and imported” with “total” as line 145-149 in the revised manuscript.

  1. Figures 3 and 4: The part B of the figures should be similarly presented. x-axis description of Fig.3B is inclined. The legend "imported cases" is redundant, as all of these are imported. Figure legends indicate "various continents" but this is misleading. Preferably "from Asian and other countries", or something similar. 

Response: The authors have revised the figure 3 and figure 4 as line 152-157 in the revised manuscript.

  1. Line 170. "The gender" is used in the middle of a sentence.

Response: The authors have revised the sentence as line 182 in the revised manuscript.

  1. Lines 202-203. campylobacteriosis and salmonellosis should not start with a capital letter. 

 Response: The authors have revised the words as line 225-226 in the revised manuscript.

Hopefully, our revised manuscript could fulfill your scientific requirements for publication.

Sincerely yours,

Chia-Peng Yu, Ph.D. (the corresponding author)

School of Public Health,

National Defense Medical Center

No.161 Sec. 6, Minquan E. Rd., Neihu Dist., Taipei 114, Taiwan, Republic of China,

Tel: +886-2-87923311 ext. 16791, Fax: +886-2-87924379,

Round 2

Reviewer 1 Report

The manuscript still needs some language editing.

Line 16: Delete "in Taiwan"

Line 24: ...(p = 0.019), with a gradual increase in the 20–40 years

Line 20: During these periods, there were two cases of....

Lines 97-102: In this study, we investigated the epidemiological features of domestic and imported cases of typhoid and paratyphoid in Taiwan during the period 2011–2020. The total number of typhoid cases was 226, the total number of domestic cases was 107, and the total number of imported cases was 119 (Table 1). For paratyphoid, total number of cases was 61, the total number of domestic cases was 19, and the total number of imported cases was 42 (Table 2).

Figures: Please remove all borders and horizontal lines form the figures.

Figure 1: I suggest a bar chart with a trend line

Figure 2: I suggest a bar chart for the total on the Primary Y axis and the line graph for the imported number of cases on the secondary.

Author Response

Dear the reviewers,                                    September 23, 2021

We resubmitted the manuscript entitled “Incidence and risk factors of notifiable typhoid and paratyphoid in Taiwan during 2011-2020” to the Journal after amendments made based on reviewers comments. We have carefully revised our manuscript according to reviewers’ critiques and suggestions (including English editing, English-Editing-Certificate-34449). We marked amendments in yellow font in the manuscript for clarity purpose. Our specific responses to reviewers’ comments are as follows.

Reviewer 1

  1. The manuscript still needs some language editing.

Response: Thanks the reviewer comment. The authors have done English editing via MDPI system ordered. Please examined to the certificate on last page (English-Editing-Certificate-34449).

  1. Line 16: Delete "in Taiwan"

Response: The authors have deleted "in Taiwan" as line 16 in revised manuscript.

  1. Line 24: ...(p = 0.019), with a gradual increase in the 20–40 years

Response: The authors have revised “...(p = 0.019), with a gradual increase in the 20–40 years " as line 24 in revised manuscript.

  1. Line 20: During these periods, there were two cases of....

Response: The authors have revised “During these periods, there were two cases of.... " as line 27 in revised manuscript.

  1. Lines 97-102: In this study, we investigated the epidemiological features of domestic and imported cases of typhoid and paratyphoid in Taiwan during the period 2011–2020. The total number of typhoid cases was 226, the total number of domestic cases was 107, and the total number of imported cases was 119 (Table 1). For paratyphoid, total number of cases was 61, the total number of domestic cases was 19, and the total number of imported cases was 42 (Table 2).

Response: The authors have revised “In this study, we investigated the epidemiological features of domestic and imported cases of typhoid and paratyphoid in Taiwan during the period 2011–2020. The total number of typhoid cases was 226, the total number of domestic cases was 107, and the total number of imported cases was 119 (Table 1). For paratyphoid, the total number of cases was 61, the total number of domestic cases was 19, and the total number of imported cases was 42 (Table 2).” as line 97-102 in revised manuscript.

  1. Figures: Please remove all borders and horizontal lines form the figures.

Response: The authors have deleted “all borders and horizontal lines form the figures ” as line141-155 in revised manuscript.

  1. Figure 1: I suggest a bar chart with a trend line

Response: The authors have revised “a bar chart with for the total on the Primary Y axis and the line graph for the imported number of cases on the secondary in figure1 as line 141-145 in revised manuscript.

  1. Figure 2: I suggest a bar chart for the total on the Primary Y axis and the line graph for the imported number of cases on the secondary.

Response: The authors have revised “a bar chart with for the total on the Primary Y axis and the line graph for the imported number of cases on the secondary in figure 2 as line 145-149 in revised manuscript.

Hopefully, our revised manuscript could fulfill your scientific requirements for publication.

Sincerely yours,

Chia-Peng Yu, Ph.D. (the corresponding author)

School of Public Health,

National Defense Medical Center

No.161 Sec. 6, Minquan E. Rd., Neihu Dist., Taipei 114, Taiwan, Republic of China,

Tel: +886-2-87923311 ext. 16791, Fax: +886-2-87924379,

Reviewer 2 Report

I think that all my points were answered and the manuscript was improved.

Author Response

Dear the reviewers,                                    September 23, 2021

We resubmitted the manuscript entitled “Incidence and risk factors of notifiable typhoid and paratyphoid in Taiwan during 2011-2020” to the Journal after amendments made based on reviewers comments. We have carefully revised our manuscript according to reviewers’ critiques and suggestions. We marked amendments in yellow font in the manuscript for clarity purpose. Our specific responses to reviewers’ comments are as follows.

Review 2

  1. I think that all my points were answered and the manuscript was improved.

Response: Thanks the reviewer comment.

Hopefully, our revised manuscript could fulfill your scientific requirements for publication.

Sincerely yours,

Chia-Peng Yu, Ph.D. (the corresponding author)

School of Public Health,

National Defense Medical Center

No.161 Sec. 6, Minquan E. Rd., Neihu Dist., Taipei 114, Taiwan, Republic of China,

Tel: +886-2-87923311 ext. 16791, Fax: +886-2-87924379,
